# The Cells Out of Sample (COOS) dataset and benchmarks for measuring out-of-sample generalization of image classifiers

**Alex X. Lu**
Computer Science,
University of Toronto
alexlu@cs.toronto.edu

**Amy X. Lu**
Computer Science,
University of Toronto
Vector Institute
amyxlu@cs.toronto.edu

**Wiebke Schormann**
Biological Sciences,
Sunnybrook Research Institute
wiebke.schormann@sri.utoronto.ca

**Marzyeh Ghassemi**
CIFAR AI Chair,
University of Toronto
Vector Institute
marzyeh@cs.toronto.edu

**David W. Andrews**
Biological Sciences,
Sunnybrook Research Institute
Biochemistry and Medical Biophysics,
University of Toronto
david.andrews@sri.utoronto.ca

**Alan M. Moses**
Cell and Systems Biology
Computer Science
CAGEF
University of Toronto
alan.moses@utoronto.ca

## Abstract

Understanding if classifiers generalize to out-of-sample datasets is a central problem in machine learning. Microscopy images provide a standardized way to measure the generalization capacity of image classifiers, as we can image the same classes of objects under increasingly divergent, but controlled factors of variation. We created a public dataset of 132,209 images of mouse cells, COOS-7 (**C**ells **O**ut **O**f **S**ample **7**-Class). COOS-7 provides a classification setting where four test datasets have increasing degrees of covariate shift: some images are random subsets of the training data, while others are from experiments reproduced months later and imaged by different instruments. We benchmarked a range of classification models using different representations, including transferred neural network features, end-to-end classification with a supervised deep CNN, and features from a self-supervised CNN. While most classifiers perform well on test datasets similar to the training dataset, all classifiers failed to generalize their performance to datasets with greater covariate shifts. These baselines highlight the challenges of covariate shifts in image data, and establish metrics for improving the generalization capacity of image classifiers.

## 1 Introduction

For a classifier to be useful predictively, it must be able to accurately label out-of-sample data (new data not seen during training). Researchers often estimate predictive performance by holding out a random subset of the training data, but this only simulates the condition where test and training data

are drawn from the same distribution. In practice, even small natural variations in data distributions can challenge the generalization capacity of classifiers: Recht *et al.* show that deep learning models trained on CIFAR or ImageNet drop in classification accuracy when evaluated on a new dataset carefully curated with the same methods as the original datasets [1], suggesting that even state-of-the-art classifiers are not robust to out-of-sample data from a more realistic setting.

While understanding the robustness of classification models to covariate shifts (situations where the distribution of out-of-sample data differs from that of training data) is broadly applicable, biomedical domains exemplify cases where the failure of image classifiers to generalize can have serious consequences. Diagnostic systems, such as those that predict pneumonia from chest radiographs, may not generalize to data from different institutions [2]. In pharmaceutical research, drugs are screened based on the effects they have on diseased cells [3]; models classifying these effects perform better on microscope images from the same sample than on reproduced experiments [4]. These issues are not exclusive to images, and are prevalent in many biomedical datasets: similar challenges include batch effects in genomics data [5], site effects in MRI data [6], and covariate shifts over time in medical records [7, 8]. Thus, validating models with realistic out-of-sample datasets is important not only for estimating performance in real use-cases where covariate shifts are unavoidable, but also for model selection: performance gains on randomly held-out test data may not translate to improvements on datasets with covariate shifts [7].

Here, we sought to create a standardized dataset for measuring the robustness of image classifiers under various degrees of covariate shift. We reasoned that microscopy experiments would allow us to image a large set of naturally variable objects (cells) under controlled factors of variation. Cells naturally vary in aspects like shape or size [9]. While still stochastic, these variations are influenced by environmental factors like temperature or humidity [10], meaning that images taken on the same day are more likely to be similar than those taken on different days or seasons. Compounding these biological variations are technical biases, such as microscope settings. Different instruments may produce subtle illumination or contrast differences, which classifiers can overfit [11, 12].

We introduce COOS-7 (Cells Out Of Sample 7-Class), a public dataset of 132,209 images of mouse cells. In addition to a training dataset of 41,456 images, COOS-7 is associated with four test datasets, representing increasingly divergent factors of variation from the training dataset: some images are random subsets of the training data, while others are from experiments reproduced months later and imaged by different instruments. We benchmark a range of classification models using different representations, both classic and state-of-the-art, and show that all methods drop significantly in classification performance on the most diverged datasets.

The full COOS-7 dataset is freely available at Zenodo (`https://zenodo.org/record/3386336`) under a CC-BY-NC 4.0 license. We provide a script to unpack all images into directories of tiff files.

## 2 COOS-7

### 2.1 Overview of images and classification setting

To create COOS-7, we curated 132,209 images of mouse cells. Each image in COOS-7 is a 64x64 pixel crop centered around a unique mouse cell. Each image contains two channels. The first channel shows a fluorescent protein that targets a specific component of the cell. Each mouse cell is stained with one of seven fluorescent proteins, which highlight distinct parts of the cell ranging from the ER to the nuclear membrane. The goal of our classification problem is to predict which fluorescent protein a cell has been stained with: Table 1 summarizes our class labels and shows example images of the first channel for each class, from three of the datasets in COOS-7.

The second channel is a fluorescent dye that stains the nucleus, consistent across all cells in our dataset. On its own, the second nucleus channel is not expected to discriminate any of the classes in our dataset, but we provide this channel to help models learn useful correlations. For example, while the Golgi (class 3) and the peroxisomes (class 5) are both characterized as bright dots in the cell, the Golgi tends to surround the nucleus, while the peroxisomes are distributed more evenly in the cell.

All images are stored in 16-bit, representing the raw intensity values acquired by the microscope, which we provide to maximize flexibility in preprocessing for methods on this dataset. For visualization purposes (and as preprocessing to the methods we benchmark), we rescale images, but users should be aware that the raw images will look different from those presented in Table 1.

Table 1: Classes and examples from COOS-7

| Label | Class | Training Examples | | | | Test3 Example | Test4 Example |
|-------|-------|---|---|---|---|---|---|
| 0 | Endoplasmic Reticulum (ER) | 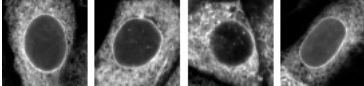 |  |  |  |  |  |
| 1 | Inner Mitochondrial Membrane (IMM) |  | | | |  |  |
| 2 | Golgi |  | | | | | |
| 3 | Peroxisomes |  | | | |  |  |
| 4 | Early Endosome |  | | | |  |  |
| 5 | Cytosol |  | | | |  |  |
| 6 | Nuclear Envelope |  | | | | |  |

## 2.2 Overview of training and test datasets

COOS-7 is curated from a larger set of microscopy experiments, spanning the course of two years. In these experiments, cells were grown on plates, containing 384 wells. Each well is a fluorescent protein, with the configuration differing from plate to plate. A robot-controlled microscope slides over the wells, taking 10-20 images for each well (see methods of [13] for a similar experimental set-up.) The original images taken by the microscope typically contain multiple cells; we process these into crops centered around individual cells by segmenting the second nucleus channel using a trained mask-RCNN, YeastSpotter [14]. We systematically imaged 95 plates, typically with about a week between plates. Four of these plates were grown and imaged at a different microscope in a collaborating institution. All images were taken using PerkinElmer OPERA® QHS spinning disk automated confocal microscope with 40x water objectives (NA=0.9).

We exploited the structure of these experiments to produce five independent datasets with different factors of variation (Table 2). We manually examined all candidate wells for our seven selected fluorescent proteins to ensure that they were free from experimental defect and contamination, and to verify the images were visually consistent with their label. With these filters, we curated a subset of high-quality images, spread over 7 plates for each class. We chose 8 wells for each plate, for a total of 56 wells per class. We roughly balanced the number of images by using a consistent number of raw microscope images per well (although there is still some class imbalance at the level of the cell crop images due to differing proliferation rates.)

Finally, we divided these plates and wells into training and test datasets, as described in Table 2. Where possible, we emphasized potential systematic biases in dividing our dataset. For Test2, we only included wells at the borders of a plate, as these wells are more susceptible to environmental effects than interior wells [15]. For Test3, we chose the two chronologically latest experiments, to emphasize potential non-stationarities over time (in most classes, the Test3 experiments differ from the Training dataset by a gap of months.)

In our classification setting, all methods must be trained and optimized using the Training dataset exclusively, and their performance evaluated on each of the four test datasets. As shown in Table 1,

Table 2: Description of datasets provided in COOS-7

| Dataset | Description | Images |
|---------|-------------|--------|
| Training | Images from 4 independent plates for each class | 41,456 |
| Test1 | Randomly held-out images from the same plates in training dataset | 10,364 |
| Test2 | Images from the same plates, but different wells than training dataset | 17,021 |
| Test3 | Images from 2 independent plates for each class, reproduced on different days than training dataset | 32,596 |
| Test4 | Images from 1 plate for each class, reproduced on different day and imaged under different microscope than training dataset | 30,772 |

while the morphology of cells and contrast of images may differ in the test datasets relative to the training datasets, their underlying classes are still visibly distinct and identifiable.

## 2.3 Related Work

Covariate shifts caused by differences in plate, well, and instruments are widely acknowledged as an issue in microscopy datasets [15] and have been demonstrated to cause classifiers on these images to overfit [11]. Previous work recognizes the importance of demonstrating robustness, and numerous methods have been proposed [4, 12, 16, 17, 18]. However, previous works have been limited in their ability to measure these effects due to lack of appropriate datasets. These studies rely on adapting pre-existing datasets, which are not designed with the purpose of measuring covariate shifts. For example, while [4] measure generalization under batch effects by holding out images from the same batch during evaluation, this procedure reduces the number of classes evaluated, as some classes are only imaged within a single batch. Similarly, high-throughput microscopy databases like [19] sometimes provide replicates of screens under the same treatment, but the arrangement of proteins on plates generally remains the same each time, prohibiting the analysis of well effects.

In contrast, our experimental design centers around randomized plate arrangements and the replication of experiments over time, directly enabling stratification of the dataset by multiple kinds of covariate shifts. Our test sets encompass covariate shifts not typically seen in other datasets: few microscopy datasets replicate experiments at different sites/microscopes. To our knowledge, COOS is the most extensive microscopy image dataset for measuring generalization under covariate shifts to date.

Compared to other datasets in computer vision, COOS resembles those used for studying domain adaptation [20]. Unlike these datasets, the test datasets in COOS should not be considered as being from a different domain than the training dataset: the datasets encompass natural variation in cells that would be difficult or impossible to control for in a realistic deployment situation. Our dataset is more similar to that of [1] in that we show that even in-domain generalization is challenging. Compared to the natural images in their dataset, we consider our cell images to be simpler. In addition, we provide multiple test datasets with controlled and known covariate shifts.

## 3 Classification Baselines for COOS-7

### 3.1 Baselines from a wide range of classifiers

To provide baselines for out-of-sample generalization on COOS-7, we extracted features and built classifiers using a variety of methods commonly used for microscopy images, both classic and state-of-the-art. Unless otherwise stated, we followed practices outlined in previous work.

First, as our classic computer vision baseline, we extracted Haralick texture features (abbreviated as Texture), which are popular for microscopy image analysis due to their rotation invariance [21].

We rescaled the intensity of each image to the range [0, 1], and extracted texture features from the first channel at 5 scales. In addition, we extracted features representing the mean, sum, and standard deviation of intensity of pixels in the first channel, and the correlation between the first and second channels.

Second, we trained a fully supervised 11-layer CNN, DeepLoc, which has achieved state-of-the-art results in classifying protein localization for 64x64 images of yeast cells [22]. We followed all preprocessing, parameterization, and model selection practices by the authors. We trained DeepLoc for 10,000 iterations with a batch size of 128 on 80% of the Training dataset, and chose the iteration (at intervals of 500) with the best performance on the remaining 20% (stratified to preserve percentage of samples per class). We report end-to-end performance, and we also extracted features from the last fully-connected layer of our trained model for building further classifiers.

Third, we extracted features from pretrained CNNs on ImageNet (abbreviated as VGG16), which has been shown to outperform classic unsupervised feature representation methods for cancer cell morphology [23]. We used a pre-trained VGG16 model, as this was the best-performing previously-reported model that would accept 64x64 images. We converted and rescaled the first channel of our images to 8-bit RGB. Contrary to the results of Pawlowski *et al.*, we observed that including features from the second channel decreased performance, so we did not include models with these features in our final baselines. We extracted features from all layers (max-pooling convolutional layers), but only report benchmarks for the top 3 overall performing layers.

Fourth, we extracted features learned from a self-supervised method designed for microscopy images (abbreviated as PCI), which has achieved unsupervised state-of-the-art results in classifying protein localization for 64x64 images of yeast cells and human cells [24]. We trained the model unsupervised on the Training dataset exclusively, following practices by the authors. We extracted features from all layers of the source cell encoder of this model (max-pooling convolutional layers), but only report benchmarks for the top 3 overall performing layers.

For each feature set, we built three classifiers on the extracted features for the Training dataset exclusively: a $k$-nearest neighbor classifier ($k = 11$), a L1 Logistic Regression classifier, and a Random Forest classifier. For all models, we centered and scaled features with the mean and standard deviation of the Training dataset. To optimize our Random Forest classifiers, we conducted a random search (100 samples) over a parameter grid (n_estimators = {20, 40, 60, 80, 120, 140, 160, 180, 200}, max_features = {'log2', 'sqrt'}, max_depth = {1, 13, 25, 37, 50, 'None'}, min_samples = {2, 5, 10, 20, 40}, min_samples_leaf = {1, 2, 4, 8, 16}), and selected the classifier with the best performance on a 5-fold cross-validation of the Training dataset. All classifiers were implemented in Python with Scikit-Learn [25].

We report the performance of all classifiers on all datasets in Table 3. We report the balanced classification error, to control for differences in class balance from dataset to dataset. The best results for each feature representation method on each test dataset are bolded.

## 3.2 All classifiers drop in performance on out-of-sample data with larger covariate shifts

All methods we tried performed well on the test datasets most similar to the Training dataset, Test1 and Test2. Features from our deep learning models had as little as 1.1% error on these datasets, but even logistic regression classifiers built on classic computer vision features achieved 6.8% error or lower. However, when attempting to generalize to the test datasets with larger covariate shifts, Test3 and Test4, all classifiers had large drops in performance (although the fully-supervised CNN achieved the lowest error on these datasets.)

We observed that all classifiers failed to generalize regardless of the complexity of the classification model. The texture features benefited more from classification models that weigh features (e.g. logistic regression versus $k$NN) compared to the models that learn features specific to a dataset (DeepLoc, PCI). Otherwise, we saw little difference in performance or generalization capacity, most exemplified by our DeepLoc results: we achieved similar error with a $k$NN classifier on the last layer's features as we did with the CNN's fully-connected classifier. These results suggest that performance and generalization is bounded by the quality of the representation, not by the complexity of the classification model.

Table 3: Class-Balanced Error (%) of Classification Models on COOS-7 Datasets

| Features | Model | Train | Test1 | Test2 | Test3 | Test4 |
|---|---|---|---|---|---|---|
| DeepLoc | End-to-End | 1.2 | 1.2 | 1.5 | 7.4 | 5.4 |
| DeepLoc (FC2) | kNN | 1.1 | 1.3 | 1.5 | 6.9 | 4.7 |
| DeepLoc (FC2) | L1 LR | 1.1 | **1.1** | **1.4** | 7.7 | **4.1** |
| DeepLoc (FC2) | RF | 0.0 | **1.1** | **1.4** | **6.8** | 5.0 |
| Texture | kNN | 10.4 | 11.8 | 11.2 | 17.6 | 25.6 |
| Texture | L1 LR | 6.4 | **6.8** | **6.5** | **12.0** | **12.1** |
| Texture | RF | 0.0 | 7.3 | 7.1 | 16.4 | 17.1 |
| PCI Conv3 | kNN | 2.2 | 2.4 | 2.7 | 8.9 | 8.0 |
| PCI Conv3 | L1 LR | 1.0 | **1.4** | **1.7** | 9.2 | 7.4 |
| PCI Conv3 | RF | 0.1 | 2.1 | 2.2 | 11.0 | 8.6 |
| PCI Conv4 | kNN | 2.1 | 2.4 | 2.5 | 10.1 | 7.8 |
| PCI Conv4 | L1 LR | 1.6 | 1.5 | 1.9 | **8.7** | 6.0 |
| PCI Conv4 | RF | 0.1 | 2.5 | 2.7 | 10.7 | 7.5 |
| PCI Conv5 | kNN | 2.6 | 2.7 | 2.9 | 12.1 | 8.9 |
| PCI Conv5 | L1 LR | 2.5 | 2.5 | 2.6 | 11.4 | **5.7** |
| PCI Conv5 | RF | 0.0 | 2.5 | 2.7 | 10.8 | 7.4 |
| VGG16 Conv3_3 | kNN | 6.8 | 7.9 | 8.2 | 12.4 | 10.0 |
| VGG16 Conv3_3 | L1 LR | 5.7 | 6.6 | 6.9 | 11.3 | 9.3 |
| VGG16 Conv3_3 | RF | 0.2 | 9.5 | 9.2 | 15.9 | 10.9 |
| VGG16 Conv4_1 | kNN | 6.5 | 7.3 | 7.6 | 9.3 | 8.4 |
| VGG16 Conv4_1 | L1 LR | 3.1 | 4.2 | 4.1 | 8.2 | **6.7** |
| VGG16 Conv4_1 | RF | 0.1 | 7.5 | 7.3 | 11.3 | 7.8 |
| VGG16 Conv4_2 | kNN | 6.6 | 7.8 | 7.8 | 9.1 | 8.4 |
| VGG16 Conv4_2 | L1 LR | 2.8 | **3.9** | **3.9** | **8.0** | 6.8 |
| VGG16 Conv4_2 | RF | 0.2 | 7.4 | 7.5 | 10.2 | 8.4 |

## 3.3 Confusion matrices reveal non-uniform errors

Next, to understand which specific classes our classifiers were failing to generalize on, we examined the confusion matrices for some classifiers on various test datasets (Supplementary Tables 1-9). We observed that covariate shifts sometimes have non-uniform effects on classification performance: in some cases, the majority of classes were predicted with very little error, with only a few classes sharply decreasing in performance.

Across the classifiers we examined, we observed that the two most common errors were classifying the early endosome as the ER or the Golgi, or the Golgi as the IMM or the peroxisomes. These errors were between the more visually similar classes in COOS-7; in contrast, the classes that were distinct from any other class in our dataset, such as the cytosol or the nuclear envelope, were generally classified well by all classifiers, across all datasets.

As an example of a case where errors were predominantly concentrated in one class, we observed for the DeepLoc classifiers on Test 4, while every other class was classified with > 0.97 sensitivity, the early endosome class was classified with only 0.684 sensitivity, compared to 0.989 sensitivity in Test1.

The non-uniform effects of covariate shifts observed here suggest that overall metrics may not always adequately describe how classifiers fail to generalize on out-of-sample data. Here, these effects are detectable due to the small number of classes, but for classification problems with many classes (such as ImageNet), a large drop in performance on only a few classes may not be detectable from metrics like the overall classification error.

### 3.4 Comparing errors between test datasets reveals variable effects of covariate shifts

In comparing confusion matrices for the same models between datasets, we observed that the types of errors classifiers made differed between datasets. For example, while all classifiers we examined had lower sensitivity on the early endosome class in both Test3 and Test4, classifiers mostly confused the early endosome with the ER in Test3, and the early endosome with the Golgi in Test4. Qualitatively, we noticed differences in the typical appearance of the early endosome class in our test datasets (as shown in the examples in Table 1), possibly due to systematic differences in morphology or microscope illumination.

We also observed that some models were robust to errors in the same classes in one dataset, but not in another. For example, the logistic regression classifier built on the VGG16 features had lower sensitivity on the Golgi class in both Test3 (0.848) and Test4 (0.866). In contrast, the logistic regression classifier built on the self-supervised (PCI) features had lower sensitivity in Test3 (0.749), but not in Test4 (0.984).

These results suggest that the exact nature of covariate shifts can differ in different out-of-sample datasets. New out-of-sample datasets may challenge classifiers in unpredictable ways, inducing errors not seen in previous out-of-sample datasets. Thus, validation on a single out-of-sample dataset may not be sufficient to conclude that a classification model is robust in general.

## 4 Conclusion

We released a new public dataset, COOS-7, specifically designed to test the generalization capacity of image classifiers under covariate shift. We demonstrated the challenge of generalizing image classifiers to out-of-sample data: no current state-of-the-art technique was able to fully compensate for covariate shifts in the datasets most different from the training data.

Our baselines highlight challenges in measuring out-of-sample generalization under covariate shift: we showed that covariate shifts will have non-uniform effects on the class-specific performance of classifiers, and that the nature of covariate shifts can differ from dataset to dataset. These observations have implications for how rigorous we need to be in validating the performance of machine learning models before deploying them into real-life applications where data distributions are not stable: overall classification metrics may understate the true effects of covariate shifts and good performance on a single out-of-sample dataset may not confirm that the model is robust to all covariate shifts in general.

We note that we intentionally designed COOS-7 to contain only visually distinct fluorescent proteins. We carefully curated a subset of higher quality experiments from a larger dataset, and approximately class-balanced the examples. These factors make COOS-7 particularly amenable to machine learning methods, and easy for most classifiers to achieve a high level of performance. Yet, even on this toy example, covariate shifts greatly hamper out-of-sample generalization. It is unknown if these covariate shifts will be exacerbated in a more realistic biological imaging setting, especially with the inclusion of even more visually similar classes. We plan to examine this problem by releasing further datasets in the future, which will include more ambiguous classes and a greater range of experimental variability.

Finally, while we focused on the value of COOS-7 for methods development in this manuscript, finding methods that improve the out-of-sample generalization baselines in this work will have practical implications for biologists working with microscopy images. Classifying protein localization is a major problem in cell biology, as a protein's localization strongly relates with its function [26]. Among other applications, accurately classifying protein localization in cells under stress or drug treatments can lead to identification of the key proteins that drive disease [27]. Since these classifiers are usually intended to automate the labeling of new data, out-of-sample generalization is essential. Here, we structured our dataset to be a protein localization classification problem: although proteins can have the same localization, we specifically chose proteins that were distinct examples of different protein localizations. We therefore expect improvements to our baselines to yield methods for more robust and generalizable prediction of protein localization, meaning that methodological work on this dataset will contribute to a more robust and generalizable understanding of protein biology.

## 5 Acknowledgements

Alex X. Lu is funded by a pre-doctoral award from NSERC. Amy X. Lu is funded by a Master's award from NSERC. Alan M. Moses holds a Tier II Canada Research Chair. Wiebke Schormann and David W. Andrews are funded by CIHR Foundation Grant FDN143312. David W. Andrews holds a Tier 1 Canada Research Chair in Membrane Biogenesis. Maryzeh Ghassemi is funded in part by Microsoft Research, a CIFAR AI Chair at the Vector Institute, a Canada Research Council Chair, and an NSERC Discovery Grant. This work was partially performed on a GPU donated by Nvidia.

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
