[Supplementary Material]

Supplementary Table 1: Confusion Matrix for DeepLoc on Test1

|   | 0 | 1 | 2 | 3 | 4 | 5 | 6 |
|---|---|---|---|---|---|---|---|
| 0 | 0.990 | 0 | 0.003 | 0 | 0.001 | 0.002 | 0.002 |
| 1 | 0.003 | 0.979 | 0.014 | 0.002 | 0 | 0.001 | 0.001 |
| 2 | 0.001 | 0.002 | 0.995 | 0.002 | 0 | 0 | 0 |
| 3 | 0 | 0.001 | 0.023 | 0.976 | 0 | 0 | 0 |
| 4 | 0.006 | 0 | 0.001 | 0 | 0.989 | 0.001 | 0.003 |
| 5 | 0.003 | 0 | 0.002 | 0 | 0 | 0.994 | 0.001 |
| 6 | 0.001 | 0.001 | 0.001 | 0.001 | 0.001 | 0 | 0.994 |

Supplementary Table 2: Confusion Matrix for DeepLoc on Test3

|   | 0 | 1 | 2 | 3 | 4 | 5 | 6 |
|---|---|---|---|---|---|---|---|
| 0 | 0.991 | 0.002 | 0.004 | 0.001 | 0.001 | 0.001 | 0.001 |
| 1 | 0.004 | 0.987 | 0.008 | 0 | 0 | 0 | 0 |
| 2 | 0.019 | 0.058 | 0.848 | 0.053 | 0.013 | 0.004 | 0.005 |
| 3 | 0 | 0.054 | 0.022 | 0.921 | 0.001 | 0.002 | 0 |
| 4 | 0.177 | 0.008 | 0.052 | 0.002 | 0.756 | 0.001 | 0.004 |
| 5 | 0.003 | 0.002 | 0.002 | 0.002 | 0 | 0.991 | 0 |
| 6 | 0.003 | 0 | 0.003 | 0 | 0 | 0.002 | 0.991 |

Supplementary Table 3: Confusion Matrix for DeepLoc on Test4

|   | 0 | 1 | 2 | 3 | 4 | 5 | 6 |
|---|---|---|---|---|---|---|---|
| 0 | 0.995 | 0.001 | 0.001 | 0 | 0.002 | 0 | 0.001 |
| 1 | 0 | 0.999 | 0.001 | 0 | 0 | 0 | 0 |
| 2 | 0 | 0.004 | 0.990 | 0.005 | 0 | 0 | 0 |
| 3 | 0 | 0.004 | 0.010 | 0.985 | 0 | 0 | 0 |
| 4 | 0.078 | 0.003 | 0.235 | 0 | 0.684 | 0.001 | 0 |
| 5 | 0.004 | 0 | 0 | 0 | 0 | 0.994 | 0.002 |
| 6 | 0.019 | 0.001 | 0.004 | 0.002 | 0 | 0 | 0.974 |

Supplementary Table 4: Confusion Matrix for Logistic Regression Classifier using PCI (Conv4) Features on Test1

|   | 0 | 1 | 2 | 3 | 4 | 5 | 6 |
|---|---|---|---|---|---|---|---|
| 0 | 0.992 | 0.002 | 0.003 | 0 | 0.002 | 0.001 | 0 |
| 1 | 0 | 0.976 | 0.018 | 0.004 | 0 | 0.001 | 0.001 |
| 2 | 0 | 0.005 | 0.981 | 0.012 | 0.001 | 0 | 0 |
| 3 | 0 | 0.010 | 0.032 | 0.958 | 0 | 0 | 0 |
| 4 | 0.002 | 0.001 | 0 | 0 | 0.993 | 0 | 0.004 |
| 5 | 0.001 | 0 | 0.003 | 0 | 0 | 0.995 | 0.001 |
| 6 | 0.001 | 0.001 | 0.001 | 0.001 | 0.001 | 0 | 0.996 |

Supplementary Table 5: Confusion Matrix for Logistic Regression Classifier using PCI (Conv4) Features on Test3

|   | 0 | 1 | 2 | 3 | 4 | 5 | 6 |
|---|---|---|---|---|---|---|---|
| 0 | 0.987 | 0.005 | 0.003 | 0 | 0.003 | 0.001 | 0.001 |
| 1 | 0 | 0.990 | 0.008 | 0.001 | 0 | 0 | 0 |
| 2 | 0.016 | 0.098 | 0.749 | 0.107 | 0.022 | 0.001 | 0.007 |
| 3 | 0 | 0.037 | 0.019 | 0.94 | 0.002 | 0.001 | 0 |
| 4 | 0.212 | 0.018 | 0.029 | 0.004 | 0.731 | 0.001 | 0 |
| 5 | 0.002 | 0 | 0 | 0.002 | 0 | 0.996 | 0 |
| 6 | 0.002 | 0 | 0.001 | 0 | 0.001 | 0.001 | 0.995 |

Supplementary Table 6: Confusion Matrix for Logistic Regression Classifier using PCI (Conv4) Features on Test4

|   | 0 | 1 | 2 | 3 | 4 | 5 | 6 |
|---|---|---|---|---|---|---|---|
| 0 | 0.992 | 0.003 | 0.001 | 0 | 0.002 | 0 | 0.002 |
| 1 | 0 | 0.999 | 0 | 0.001 | 0 | 0 | 0 |
| 2 | 0 | 0.002 | 0.984 | 0.013 | 0 | 0 | 0 |
| 3 | 0 | 0.005 | 0.027 | 0.968 | 0 | 0 | 0 |
| 4 | 0.068 | 0.004 | 0.261 | 0 | 0.647 | 0 | 0.019 |
| 5 | 0 | 0 | 0.001 | 0 | 0 | 0.999 | 0 |
| 6 | 0.002 | 0.001 | 0.002 | 0.001 | 0 | 0.002 | 0.991 |

Supplementary Table 7: Confusion Matrix for Logistic Regression Classifier using VGG16 (Conv4_2) Features on Test1

|   | 0 | 1 | 2 | 3 | 4 | 5 | 6 |
|---|---|---|---|---|---|---|---|
| 0 | 0.969 | 0.006 | 0.004 | 0 | 0.016 | 0.003 | 0.002 |
| 1 | 0.003 | 0.944 | 0.048 | 0.003 | 0 | 0 | 0.001 |
| 2 | 0.002 | 0.017 | 0.938 | 0 | 0 | 0 | 0 |
| 3 | 0 | 0.014 | 0.047 | 0.939 | 0 | 0 | 0 |
| 4 | 0.029 | 0.007 | 0.006 | 0.002 | 0.954 | 0.001 | 0.002 |
| 5 | 0.002 | 0.003 | 0.002 | 0.001 | 0.002 | 0.991 | 0 |
| 6 | 0 | 0.003 | 0.001 | 0 | 0.001 | 0.001 | 0.993 |

Supplementary Table 8: Confusion Matrix for Logistic Regression Classifier using VGG16 (Conv4_2) Features on Test3

|   | 0 | 1 | 2 | 3 | 4 | 5 | 6 |
|---|---|---|---|---|---|---|---|
| 0 | 0.967 | 0.011 | 0.006 | 0 | 0.013 | 0.002 | 0.001 |
| 1 | 0.007 | 0.977 | 0.009 | 0.002 | 0.005 | 0 | 0 |
| 2 | 0.008 | 0.036 | 0.848 | 0.082 | 0.014 | 0.007 | 0.004 |
| 3 | 0 | 0.011 | 0.083 | 0.900 | 0.001 | 0.005 | 0.001 |
| 4 | 0.163 | 0.007 | 0.043 | 0.004 | 0.778 | 0.001 | 0.003 |
| 5 | 0.004 | 0.001 | 0.003 | 0.002 | 0.005 | 0.984 | 0.001 |
| 6 | 0.003 | 0.002 | 0.003 | 0 | 0.003 | 0.003 | 0.987 |

Supplementary Table 9: Confusion Matrix for Logistic Regression Classifier using VGG16 (Conv4_2) Features on Test4

|   | 0 | 1 | 2 | 3 | 4 | 5 | 6 |
|---|---|---|---|---|---|---|---|
| 0 | 0.979 | 0.005 | 0.004 | 0 | 0.008 | 0 | 0.004 |
| 1 | 0.008 | 0.988 | 0.002 | 0.001 | 0.001 | 0 | 0 |
| 2 | 0 | 0.043 | 0.866 | 0.089 | 0 | 0.001 | 0 |
| 3 | 0 | 0.007 | 0.026 | 0.967 | 0 | 0 | 0 |
| 4 | 0.078 | 0.018 | 0.166 | 0.001 | 0.736 | 0.001 | 0 |
| 5 | 0.002 | 0 | 0 | 0 | 0 | 0.997 | 0 |
| 6 | 0.004 | 0.001 | 0.001 | 0 | 0.004 | 0.003 | 0.989 |