[Reviews · NeurIPS 2019]

Reviewer 1



---- Summary ---- The authors contribute a dataset of microscopy images to benchmark image classifier generalization. The dataset contains tens of thousands of 64x64 pixel images of cells. Each image shows cells labelled with a fluorescent protein. In total, seven different fluorescent proteins were employed. The authors consider the image analysis task of classifying images w.r.t. these seven fluorescent labels. The data is divided into a number of sub-sets according to properties that are hypothesized to yield significant covariate shifts, like acquisition day, acquisition site and microscope, and position of the imaged cells on the imaged grid of wells on a plate (center vs fringes). The authors quantitatively evaluate classical and state of the art methods in terms of generalization from a training set (drawn from all sub-sets) to the individual subsets. They show that the performance of all evaluated methods drops considerably for all kinds of transfer. Furthermore, they show that a drop in performance when transferring to one subset is not necessarily indicative of the performance when transferred to another subset. ---- Comments ---- The paper is easy to follow as it is clearly written and well-organized. The authors contribute a novel, large dataset of images and benchmark known methods for image classification. The dataset appears to be immensely useful for method development / benchmarking of transfer learning techniques. That said, it did not become fully clear to me what the added benefit of the proposed dataset is, compared to the cited related work from biology as well as computer vision. More images? Known reasons for covariate shift? Readily available as benchmark? The authors exclusively consider the case of training on a training set (supervised or unsupervised) and then evaluating on their test sets. They do not evaluate any method for unsupervised transfer learning on the test sets. There are probably good reasons for limiting the evaluation this way. A respective discussion would improve clarity. Furthermore, the relevance of the considered classification task for applications in Biology is not discussed. I am not aware of an application where the fluorescent stain used for sample prep is unknown after imaging and has to be classified automatically. Is the dataset intended purely for technical method development, or is there a related application in Biology? Please discuss. --- Post Author Feedback --- The author feedback clarified that the biological application they consider is protein *localization*, and not protein classification. The latter would be a toy task because labelled proteins would be known a priori. The proteins used in the proposed dataset have known localizations, and hence in this case the classification task coincides with the localization task. If I understand correctly, the idea is that proteins different from the 7 used in their dataset, with varying localizations indicative of cell function and stress, would be labelled in "real" application data. Is protein localization always unique in these cases? Isn't transfer to such "real" data potentially much harder than within the different test sets of the studied data? In summary, I still don't fully understand to which extent the presented data and classification task is a "toy task" w.r.t. the targeted application of protein localization.

Reviewer 2



This paper proposed a good motivation on the issues of existing out-of-sample detection work, and then proposed a new dataset that helps to evaluate the quality of out-of-sample results. Based on this new dataset, the paper further conducted an empirical study of existing baselines and showed the challenges of solving the out-of-sample problem. The value of this work mainly lies in the creation of the new dataset that helps to address the out-of-sample detection problems.

Reviewer 3



This works provides an original, high quality, and clearly explained contribution to the field of machine learning - in the form of a valuable dataset. I expect this dataset to have a significant impact on the field perhaps becoming a new benchmark.

[Author Response · NeurIPS 2019]

We thank the reviewers for their supportive comments. In particular, we are glad that the reviewers state that our work is an "original, high-quality, and clearly explained contribution to the field of machine learning" that "facilitates future methodological contributions" by providing a dataset that is expected to "have a significant impact on the field perhaps becoming a new benchmark". We have addressed questions and comments below.

**Reviewer 1: Please discuss what precisely is the added benefit of the presented dataset over the data used in the cited related work on biology applications as well as computer vision.**

To our knowledge, COOS is the most extensive biological image dataset for measuring generalization under covariate shifts to date. Previous work recognizes the importance of demonstrating robustness, but have been limited in their ability to measure these effects. These studies rely on adapting pre-existing datasets, which are not designed with the purpose of measuring covariate shifts. For example, while [1] measure generalization under batch effects by holding out experimental batches, this procedure reduces the number of classes evaluated, as some classes are only imaged within a single batch. Similarly, while [2] provides replicates of yeast microscopy screens under the same treatment, the arrangement of proteins on plates remains the same each time, prohibiting the analysis of well effects.

In contrast, our experimental design centers around randomized plate arrangements and the replication of experiments over time, directly enabling stratification of the dataset by multiple kinds of covariate shifts. Our test sets encompass covariate shifts not typically seen in other datasets: few microscopy datasets replicate experiments at different sites/microscopes. Finally, we organized, preprocessed, and balanced the test sets around this metadata. Although these procedures could be performed on other microscopy datasets, they are non-trivial for researchers not familiar with the domain. Our standardized archives therefore make these images more accessible to the machine learning community.

While many other computer vision datasets focus on domain adaptation, ours is more similar to [3] in that we show that even in-domain generalization is challenging, although we provide a simpler problem, with controlled and known covariate shifts. If accepted, we will update the introduction in the camera-ready version to better contextualize our contributions with previous image datasets.

**Reviewer 1: Please discuss the relevance of the considered classification task for applications in biology.**

Classifying protein localization is a major problem in cell biology, as a protein's localization strongly relates with its function [4]. Along other applications, accurately classifying protein location in cells under stress or drug treatments can lead to identification of the key proteins that drive disease [5]. In our dataset, we specifically chose proteins that were distinct examples of the different ways proteins can localize in a cell in general, making it a representative test case for how well these classifiers generalize under the typical covariate shifts present in microscopy experiments. We expect that improving our baselines will yield methods for more robust and generalizable prediction of protein localization.

While we focused on the uses of COOS for method development, we agree that researchers developing methods would appreciate an overview of their biological contributions. We will update the discussion in the camera-ready version.

**Reviewer 1: They do not evaluate any method for unsupervised transfer learning on the test sets. There are probably good reasons for limiting the evaluation this way. A respective discussion would improve clarity.**

We agree that there is a wide range of possible strategies on this dataset, unsupervised transfer learning among them, and will highlight some of these options for future work in the discussion of the camera-ready version.

**Reviewer 3: Deposit the dataset in a repository. Provide details of the license under which the dataset is being distributed.**

We have deposited our dataset in Zenodo, under a CC-BY-NC 4.0 license. In the camera-ready version, we will update the manuscript to provide a link and details on the license.

**Reviewer 3: Provide more details on the imaging methods. Redo table 3 to use colors or bars to represent differences among different classifiers. Reference similar class of problems in genomics and MRI.**

We will update the camera-ready version of our manuscript to incorporate these details.

[1] D. Michael Ando et al. Improving Phenotypic Measurements in High-Content Imaging Screens. *bioRxiv*, Jul 2017.
[2] Judice L Y Koh et al. CYCLoPs: A Comprehensive Database Constructed from Automated Analysis of Protein Abundance and Subcellular Localization Patterns in Saccharomyces cerevisiae. *G3*, 5(6):1223–32, Jun 2015.
[3] Benjamin Recht et al. Do ImageNet Classifiers Generalize to ImageNet? *ICML*, Feb 2019.
[4] Ying-Ying Xu et al. Bioimage-based protein subcellular location prediction: a comprehensive review. *Frontiers of Computer Science*, 12(1):26–39, Feb 2018.
[5] Mien-Chie Hung and Wolfgang Link. Protein localization in disease and therapy. *Journal of cell science*, 124(Pt 20):3381–92, oct 2011.


[Meta-Review · NeurIPS 2019]

The main strength of this paper is that it proposes an interesting new dataset of biological images, which has the potential to become an interesting benchmark particularly for problems with realistic "covariate shift" problems. The problem of generalization to the same task with differences in data acquisition protocols is a big issue in empirical sciences - especially in genomics and radiology. Changes in acquisition protocol, software updates or differences in hardware can throw off a well-performing classifier. This dataset will help tackle this problem. There was a consensus among reviewers that creating such a benchmark (and testing the performance of baseline methods) is an important effort that would be of interest to the NeurIPS community. The relevance of the benchmark to solve an important real-world problem (prediction of protein localization) was less evident.